# A semi-structured questionnaire survey of laboratory animal rehoming practice across 41 UK animal research facilities

**Tess Skidmore** *, **Emma Roe**

School of Geography and Environmental Science, University of Southampton, Southampton, England, United Kingdom

* t.a.skidmore@soton.ac.uk

## Abstract

If a laboratory animal survives an experiment without lasting compromised welfare, its future must be negotiated. Rehoming may be a consideration. This paper reports on research findings that provide an indication of the uptake of animal rehoming by UK facilities and the associated moral, ethical, practical and regulatory considerations that inform decisions to rehome or not. This research addresses a widely acknowledged gap in the literature to understand both the numbers, and types of animals rehomed from UK research facilities, as well as the main motivations for engaging in the practice, and the barriers for those facilities not currently rehoming. From the ~160 UK research facilities in the UK, 41 facilities completed the questionnaire, giving a response rate of approximately 25%. Results suggest rehoming occurs routinely, yet the numbers are small; just 2322 animals are known to have been rehomed between 2015–2017. At least 1 in 10 facilities are rehoming. There exists a clear preference for the rehoming of some species (mainly cats, dogs and horses) over others (rodents, agricultural animals and primates). Indeed, although 94.15% of species kept in laboratories are rodents, they make up under a fifth (19.14%) of all animals known to be rehomed between 2015–2017. The primary motivation for rehoming is to boost staff morale and promote a positive ethical profile for the facility. Barriers include concern for the animal's welfare following rehoming, high scientific demand for animals that leaves few to be rehomed, and, finally, certain animals (mainly those genetically modified) are simply unsuited to rehoming. The findings of this research will support facilities choosing to rehome, as well as those that are not currently engaging in the practice. By promoting the practice, the benefits to rehoming in terms of improving laboratory animal's quality of life, helping facility staff to overcome the moral stress of killing, and addressing public concern regarding the fate of laboratory animals, can be attained. It is only once an understanding of rehoming from the perspective of UK research facilities has been ascertained, that appropriate policy and support can be provided.

## Introduction

Rehoming is defined by the UK Home Office (pg. 10) as "*the movement of a relevant protected animal from an establishment to any other place that is not an establishment under A(SP)A.*"

**Data Availability Statement:** There are ethical restrictions on sharing the data gathered from the questionnaire. Data cannot be shared resulting from the impossibility of fully anonymising

questionnaire results. To allow access to the data through sharing is in direct contradiction to the ethical agreement signed by participants, and approved by the University of Southampton's ethical committee. The University of Southampton's ethical committee (Head of Research Governance) can be contacted at: rgoinfo@soton.ac.uk.

**Funding:** T.S and E.R 205393/D/16/Z Wellcome Trust https://wellcome.ac.uk/ The funders had no role in study design, data collection and analysis, decision to publish, or preparation of the manuscript.

**Competing interests:** The authors have declared that no competing interests exist.

The "place" referenced is most commonly a farm, aquarium, zoo or private home [1]. Directive 2010/63/EU states that animals can be rehomed if: "the state of the health of the animal allows it", "there is no danger to public health, animal health, or the environment", and if "appropriate measures have been taken to safeguard the well-being of the animal"[2]. The Directive makes explicit that those animals whose welfare would be compromised if rehomed should be killed at the end of experiments [2].

The practice of rehoming is guided by the notion that animals are sentient beings and worthy not only of avoiding suffering, but also of experiencing a good quality of life [3]. Despite laboratory animals' role in important medical advances, the use of animals in scientific research remains a controversial issue [4]. Rehoming addresses the arguably unnecessary killing of some animals after being used in a scientific procedure. The killing of research animals is undertaken for three primary reasons– 1) as a scientific requirement, 2) to prevent avoidable suffering (euthanasia), or 3) for financial/logistical reasons [5]. As will be discussed, the main opportunity for rehoming lies where humane killing would otherwise take place for financial or logistical reasons.

'Surplus' or "bred but not used" [6] animals are often cited as the most appropriate candidates for rehoming [5] for the following reasons. Firstly, surplus animals have not been subject to research, and thus long-term health implications (often cited as a barrier to rehoming) are less likely to represent a risk [7]. Secondly, rehoming these animals typically presents a lower risk of disease transmission [5]. Thirdly, the "moral stress" [8] experienced by those associated with the killing of these animals tends to be higher for surplus animals because of a cultural awareness of ideas of waste. Killing these animals can be perceived as "wrong"—especially when the animal could be placed in a potential home [5, 9]. Thus, rehoming laboratory animals may benefit staff morale [10].

The reasons to consider rehoming are not only proposed by those working inside the laboratory; public opinion should also be taken into account. Killing, even when undertaken humanely, can evoke strong negative emotions in the public [11]. Some may view it as an infringement on the right to life, underrating its inherent value [5]. Euthanasia may have more positive connotations in the veterinary clinic setting when companion animals are relieved of suffering, but in the animal laboratory, issues with routine killing are compounded within a setting where animals are systemically harmed for human benefit. Thus, research should seek to address societal concerns and reflect these concerns appropriately within policy guiding animal research.

Indeed, despite the widely recognised benefits to rehoming, and concerns surrounding the routine nature of killing within animal research facilities, no literature quantifies how many animals are rehomed from UK laboratories. However, existing research speculates that it is likely to be undertaken with very small numbers of animals [5], and thus currently constitutes the exception rather than the rule. There is a lack of research that explores the extent of laboratory animal rehoming practices at a national level, nor is there much detail regarding the process of laboratory animal rehoming when it does occur.

## Rehoming and animal welfare

Attaining high standards of animal welfare in the farm, zoo, laboratory, and for companion animals is an important societal concern [12]. Animal welfare typically includes a consideration of animal's affective state (pain), biological functioning (injuries), and sometimes also a consideration of naturalness (such as pasture access) [13]. Novel and innovative approaches for achieving good animal welfare are increasingly considered. The rehoming of laboratory animals represents one way in which attempts could be realised, as the registers of ethical

concern about practising good animal welfare shift towards the case for more rehoming. This section will examine how rehoming represents both an opportunity to improve, but equally compromise, animal welfare.

The Farm Animal Welfare Committee (FAWC) proposed the notion of "a life worth living" [14]. A life worth living means that the balance between negative and positive experiences is favourable, and is achieved by complying with minimum welfare standards coupled with the promotion of positive experiences [15]. It includes the degree to which the animal is provided with its needs and wants, resulting in good health and happiness, and also longevity [16]. Longevity in the context of rehoming is important, as the practice could help to extend animal lives as well as promoting positive life experiences. If an animal would have a life worth living, then death is contrary to the animal's individual interests, as it involves the absence of positive states [17]. Rehoming represents a way to enhance quality of life, and extend life. Hence, research should examine the processes by which it occurs in order to understand best practice, improve policy and promote rehoming as an option once research is concluded. The voice of researchers is valuable here: "Don't we, as researchers, owe our animals a different life after they have completed their contributions to science?" (pg. 506) [18].

However, it is also crucial to acknowledge that rehoming is not always in the best interests of the animal, and may instead serve to compromise welfare. Current UK regulatory guidance, such as the Animals in Scientific Procedures Act (1986), or A(SP)A, helps to ensure standards of welfare are maintained in the laboratory, yet once rehomed this legislation is no longer in place to protect the animal in question. It is thus the responsibility of the facility to ensure that rehoming will in no way compromise welfare [19]. This must remain an absolute priority. Factors that can affect welfare during the rehoming process are the animal's state of health, the duration and condition of transport to the new home, and the social and/or physical environment the animal will be moved to [7]. Research notes that potential compromises to welfare can be minimised or eradicated with careful and thorough planning of key processes such as transport [7] and socialisation [9, 19] to lower animal stress.

## Existing research on laboratory animal rehoming

There is a lack of academic literature focusing specifically on the politics, ethics and practices of laboratory animal rehoming, and that which has been undertaken focuses on the success of rehoming practice. This includes research on the rehoming of cats [20], dogs [19, 21, 22], ferrets [23] and primates [7, 18]. Research shows beagles have a high adaptive capacity, fit in well with families and thus make good companion animals [21]. Within 6–12 weeks of rehoming, behaviour tests on the dogs reflected relaxed body language, as well as reduced heart rates, signifying calm behaviour once settled in the home environment [21]. Research also found cats were successfully rehomed from a research facility, with a retention rate of 93.5% [20]. Of those rehomed, 80.4% were considered a valued family member, and behavioural problems were reported in just 11.3% of the cats within 6 months of adoption [20]. Finally, a study revealed that Cornell University have developed a successful direct adoption scheme, and that the University of California, San Francisco employs an indirect scheme, whereby animals are transported to a third party shelter before rehoming [9]. Both schemes were judged to be successful, with "hundreds" of animals placed in adoptive homes [9]. These case studies convey hope that rehoming as a practice could be successfully institutionalised and adapted to specific facility needs, across a range of sentient species.

However, the overwhelming focus on large mammal species does not provide a comprehensive understanding of all rehoming practice across species. Although companion animals are more commonly considered for rehoming, the practice occurs across many

species—including rabbits, rats, guinea pigs and even mice [18]. In fact, the "small size, easy and affordable maintenance and short longevity" (pg. 197) of these animals may reduce the level of commitment needed from potential adopters [5]. Despite this, there has been little work undertaken focusing on rodents, fish or agricultural animals (all of which are commonly used in animal research), and consequently there is a need to expand understandings of rehoming frequency for these species, as well as the channels by which this occurs. This is especially true as the majority of facilities keep rodents [24–26], so providing information to support them could increase rehoming prevalence.

**Purpose and significance of study.**   Although existing work evaluates rehoming in specific circumstances, no research has been undertaken to gauge the current situation regarding the numbers rehomed from UK laboratories, and which species are more commonly considered. Existing literature notes this as an essential next step in understanding the rehoming process [9, 27]. This is necessary to enhance animal welfare by following correct rehoming procedures which can help to ensure animal welfare after rehoming, as stipulated by law, and promote the rehoming of laboratory animals where viable. Most of the existing literature is based upon case studies, and focuses on larger mammal species. Finally, although research has identified motivations to rehome and reasons why facilities are not engaging in the practice, no work has quantified which motivations and barriers are considered to be the most important by the research facilities themselves. This is necessary to adapt existing policy accordingly, and support facilities if they choose to rehome their animals in the future.

As such, the key research questions are:

- How many UK facilities are known to be rehoming?

- How many animals, and what species, are being rehomed?

- What are the motivations for rehoming, and the barriers for those not currently participating in the practice?

- What range of activities does the rehoming process typically involve?

- What are the main reasons facilities identify for not being able to rehome?

## Methodology

Engagement in, and perceptions of, rehoming were measured using a specially designed questionnaire. Reponses were collected between July 2018 –January 2019. The questionnaire was designed using the University of Southampton's software, iSurvey, a survey generation tool allowing the dispersion of online surveys. The questionnaire was split into 6 sections: 1) Role and background both of the respondent and the facility they represented, 2) The facilities' rehoming policy, 3) Barriers to rehoming, 4) Opportunities presented by rehoming, 5) The rehoming process, and 6) Reasons for choosing not to rehome animals (S1 Appendix). Logic questions were applied, so the survey format varied between participants, and respondents were only presented with questions relevant to them based upon their previous answers. The survey included both closed and open questions, but comprised mostly of checkbox options, allowing it to be completed quickly and easily. In the case a suggested option was not relevant, participants were able to select an 'other' box and manually add in their response. Enabling the survey to be completed online increased convenience for the participant, as well as enabling more efficient distribution. The majority of participants completed the questionnaire within 10 minutes.

Given the sensitive nature of the research and difficulty in contacting participants (staff working at animal research facilities), they were approached indirectly through the auspices of the Animals in Science Committee and the AWERB (Animal Welfare Ethical Review Body) Hub network. It was important that respondents were able to participate on behalf of their facility, because the questions were assessing views at the facility rather than the personal level. Thus, any employee could complete the survey if they had the necessary data/knowledge. Reminder emails were circulated twice to increase participation. 41 facilities out of the ~160 UK research facilities currently operating in the UK completed the survey—giving a response rate of approximately 25%. Incomplete surveys were not counted. Nvivo12 was used to analyse the qualitative responses, Microsoft Excel (2016) to analyse quantitative findings and assist in numerical analysis (including calculating numbers of animals rehomed), and SigmaPlot (Version 13) to enable graph production.

In order to calculate the numbers of animals kept in UK research facilities to enable a comparison to the numbers rehomed, the "total animals used for the first time in experimental procedures"[26] was used. As GA (genetically altered) animals cannot be legally rehomed, the "creation & breeding of GA animals not used in experimental procedures" was omitted from the analysis. To calculate the numbers of surplus animals—which research reports are the most common rehoming candidates—the Home Office document titled "Additional statistics on breeding and genotyping of animals for scientific procedures, Great Britain 2017" was used. This is because it includes, for the first time since reporting began, non-GA animals that were bred for scientific procedures but were killed or died without being used in such procedures. However, it only states the number not used (1.81 million animals) and attributes 80% of the figure to mice, 11% rats and 7% fish. In order to calculate the remaining 2% of 'other' animals, we employed a weighting system whereby the same ratios of animals used for the first time in procedures were applied to the remaining 2% of animals (here cats, dogs other than beagles, beagles, primates, horses, rabbits, guinea pigs, gerbils, hamsters, ferrets, birds, quail, goats, sheep, cattle, pigs and amphibians). As these figures are not available for the years 2015 and 2016, we multiplied the figure by 3 in order to get an average across the 2015–2017 reporting years.

Analysing data from the open questions involved a structured inductive thematic analysis. This helped to identify common topics, ideas, concepts and patterns arising from the qualitative, open answer data. In order to do this, we used Nvivo12 to code the data and to generate themes from it. Using Nvivo12, the thematic analysis also entailed a frequency count, whereby it was possible to see the number of times each identified theme was referenced across all participants.

The questionnaire was considered and approved by the University of Southampton's Ethics Committee. All those who responded to the questionnaire were provided with a participant information sheet, and were given the opportunity to ask questions about the study. Participants provided their consent to participate in the research before completing the questionnaire. Other than protection of personal data and the anonymisation of results, the research was not considered to raise significant ethical issues.

## Results and discussion

### Context

Participants represented a variety of roles, including but not limited to: Establishment Licence Holders (ELHs), Named Veterinary Surgeons (NVSs), AWERB chairs, Named Animal Care and Welfare Officers (NACWOs), and Named Information Officers (NIOs). In terms of the species kept at facilities, the questionnaire results reflect accurately the wider landscape of UK

research institutions [24–26]. The majority of facilities that completed the questionnaire had mice (36 facilities), rats (29 facilities), and fish (23 facilities). A small number of facilities kept dogs (6 facilities), primates (4 facilities), horses (4 facilities) and cats (2 facilities). The types of research undertaken at the facilities also varied, including but not limited to; conservation, human medicine development, teaching, and animal behaviour, welfare and nutrition. Both public and private facilities completed the questionnaire.

## Number of facilities participating in rehoming

As there are ~160 UK animal research facilities, the 19 facilities that the survey found to be rehoming constitutes approximately 11% of the total number of UK research establishments (Fig 1). It is possible to say with certainty that at least 11.9% of UK facilities are rehoming, and that at least 13.8% have not engaged in the practice from the 2015–2017 period, but scaling up to give speculative figures for the whole sector is not possible. The questionnaire indicates that rehoming is considered as a possibility in UK research facilities. The fact rehoming is often a consideration in UK research facilities is also demonstrated by only one facility of the 41 facilities (which kept solely fish) suggesting that they were "*unaware that rehoming was possible*" from the closed answer questions.

## Comparing numbers of animals kept to those rehomed

Numbers rehomed across the 19 facilities from the years 2015–2017 are very low—just 2322 animals were rehomed (Table 1). Both consideration for rehoming and the numbers rehomed appears to depend heavily upon the species in question. Those species kept in larger numbers (such as mice, rats and fish) are less likely to be rehomed, whilst those kept in smaller numbers, such as cats and dogs, are more likely to be considered (goats and quail provide notable exceptions). Birds are also rehomed in large numbers, although they are also kept in higher numbers within facilities. Despite 10,141 primates being used in research, none of the four facilities that completed the survey and kept primates had rehomed them in the last 3 years. However, it should be noted that primates can be retired from research, whilst still living at the facility.

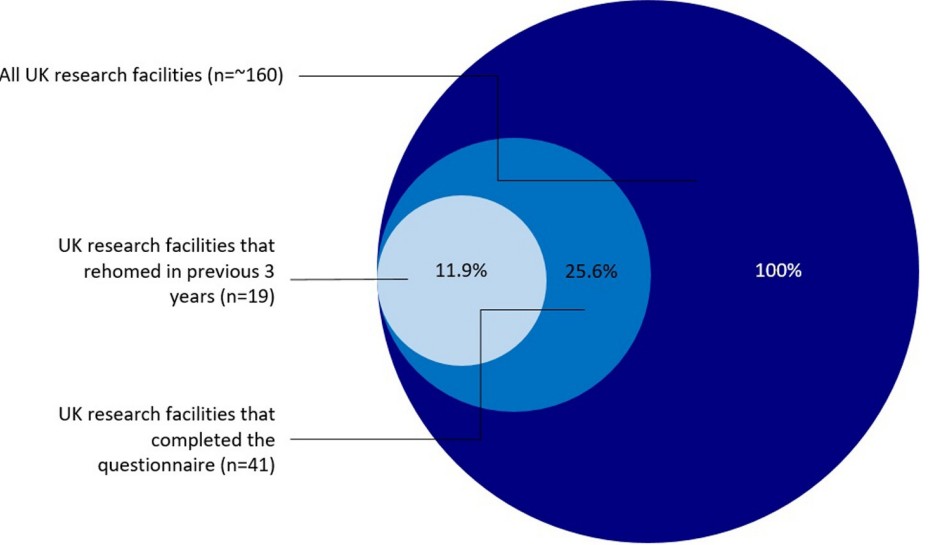

**Fig 1. UK research facilities that have rehomed in 2015–2017 period and completed the questionnaire as a percentage of all UK research facilities.**

**Table 1. A comparison of numbers of animals kept (using Home Office statistics) from 2015–2017, and numbers known to be rehomed (from 41 facilities that completed the survey).** The colour coding helps to show which animal groups are kept in higher numbers within facilities, and which are rehomed in higher numbers. Higher numbers are represented in more saturated colours.

| | Numbers kept (first time use in procedure and bred but not used) 2015–2017 across all UK facilities | Numbers rehomed of those that completed the questionnaire (n = 41 of the ~160 UK facilities) (2015–2017) |
|---|---|---|
| Cats | 448 | 171 |
| Dogs other than beagles | 447 | 71 |
| Horses | 1406 | 69 |
| Gerbils | 943 | 19 |
| Cattle | 10580 | 64 |
| Beagles | 10456 | 44 |
| Hamsters | 4742 | 16 |
| Ferrets | 1746 | 4 |
| Amphibians | 14706 | 31 |
| Fish | 1266584 | 1277 |
| Birds | 495889 | 383 |
| Rabbits | 41080 | 18 |
| Pigs | 17211 | 5 |
| Guinea pigs | 83886 | 18 |
| Rats | 1317886 | 103 |
| Sheep | 141941 | 7 |
| Mice | 7912669 | 22 |
| Goats | 726 | 0 |
| Primates | 8196 | 0 |
| Quail | 37 | 0 |

There is academic, public and policy acknowledgment of higher levels of sentience in primates [28], and in America, chimpanzees are legally entitled to retirement [29]. Research advocates that, in the US, chimpanzees should be retired as a moral imperative which acknowledges claims for redress against "histories of displacement, confinement and experimentation" (pg. 619) [29], yet in the UK context there exist worries regarding the welfare of rehomed primates. For example, one facility explained they did not currently engage in rehoming due to the difficulty in maintaining primate social groups established in the laboratory, and consequently the negative welfare implications of separation when undertaking rehoming.

Although 94.15% of species kept in laboratories are rodents, they make up under a fifth (19.14%) of all animals known to be rehomed between 2015–2017. Conversely, birds, cats, dogs, horses, amphibians and agricultural animals constitute 80.86% of total species rehomed, despite making up just 5.84% of those kept (*see* Table 1 *for more details*). This is based on the following grouping: dogs (beagles and all other dog breeds), small mammals (rats, mice, gerbils, rabbits, hamsters, ferrets, guinea pigs), birds (common quail and all other birds), agricultural animals (cattle, sheep, pigs and goats), and cats, horses, amphibians and primates. Fish were excluded from this analysis due to one outlier facility having rehomed over 1200. There thus exists a preference for the rehoming of some species over others.

## The rehoming process

**Preparation of the animal.** This section reports on qualitative (semi-structured questions) and quantitative (closed questions) findings from the survey. Selecting the most suitable

Table 2. Animal suitability for rehoming frequency count (n = 19).

| Factors to consider as raised by participants | Number of times referenced |
|---|---|
| Health | 14 |
| Temperament | 8 |
| Age | 3 |
| Breed | 1 |
| Species | 1 |

animals for rehoming was deemed an important and thorough process by the participants, commonly undertaken following facility-wide rehoming policy (14 of the 19 facilities that rehomed followed their own rehoming policy). The survey revealed that various factors were considered to be significant when assessing the suitability of an animal for rehoming. These included the animal's health, their age, breed, species and temperament, as well as the procedures they had undergone which would dictate their long-term health. Table 2 (below) provides a frequency count of these aspects as referenced across all participants.

The majority of respondents (14 of the 19 facilities that rehomed) referenced the importance of the Named Veterinary Surgeon (NVS) and relevant heath checks in this process, who is typically in charge of judging and enabling a comprehensive assessment of overall welfare and quality of life post-rehoming. The responses collected suggest that the most important factor to consider is the health of the animal, and if that cannot be guaranteed, then rehoming should not be attempted. As one respondent, a Named Information Officer at a facility that kept amphibians, birds and rodents, wrote: the *"NVS and the NACWO must confirm that the condition of the animal and its health allows for rehoming."*

In terms of preparation for rehoming, larger mammal species typically required greater effort from laboratory staff (in terms of time and resources) to rehome. This commonly included establishing and completing comprehensive and effective socialisation and training schemes, ensuring exposure to new environments, and undertaking necessary medical procedures (such as neutering). All four stages of preparation were noted to be necessary both for rodents, and for cats and dogs. These were not considered necessary with livestock (only socialisation deemed appropriate) and no preparation was required for the rehoming of fish and amphibians. Despite the fact it would seem that rehoming larger mammals required greater effort in terms of time and cost, this does not appear to hinder efforts to rehome them, and rehomed numbers are still much higher in these species.

**Finding the right home.** The majority of animals were rehomed to staff, or their friends and family (18 facilities out of the 19 that rehomed employed this pathway to find homes). This route was more commonly sought out when rehoming smaller numbers of rodents. Rehoming this way was beneficial in that owner preparation was unnecessary as those rehoming the animals were facility employees (commonly animal technicians) who already had experience with the species in question. Eight facilities used word of mouth to find homes, suggesting there is an acceptance of rehoming to the public, but that this is not generally advertised openly. Two facilities transferred their animals to third party rehoming organisations that undertook the rehoming process on their behalf, signifying that effective partnerships can be forged between facilities and rehoming organisations. Rehoming did not necessarily entail rehoming to private family homes; homes were also found in bird breeders, animal sanctuaries, schools, farms, and petting zoos.

Evaluating owner capability was judged very important in the rehoming process– 16 of the 19 facilities that rehomed required the prospective owner to meet certain criteria. An

Table 3. Owner criteria frequency count (n = 19).

| Factors to consider as raised by participants | Number of times referenced |
|---|---|
| Prior species experience | 11 |
| Suitable housing | 3 |
| Owner questionnaire completion | 2 |
| Home inspection | 2 |
| Demonstrate handling ability | 1 |

evaluation of owner capability is also important in animal shelters; owner questionnaires, interviews with shelter staff, and home visits to establish levels of knowledge regarding pet behaviour and physiology are recommended [30, 31]. This is important as owner lifestyle and circumstances greatly affects the quality of life of companion animals [32, 33]. In this research, much emphasis was placed not only on finding a home, but also on ensuring it was the 'right' home for the animal. This form of matching is not specific to laboratory animals; literature in the animal shelter context finds it equally important [34]. Criteria included that the prospective owner must be able to demonstrate that they have previous species experience, suitable housing, and handling ability (Table 3). In addition to this, and mainly for species such as horses, cats and dogs, the NVS may visit/inspect the proposed home to ensure its suitability. The potential new owner may have to complete a questionnaire (two facilities required this of new owners), which includes questions investigating the motivations to rehome the animal, previous experience of owning an animal of the same species, as well as an enquiry into the personal situation (other animals or children in the home, rural/urban environment, current employment).

As well as having selection criteria, once the new owner had been identified questionnaire responses suggested that preparing the new owner was also important (12 facilities out of the 19 that rehomed undertook some form of owner preparation). This preparation was wide ranging (Table 4), and included that contact with the NVS should be sought if any medical issues arise in the future. Some facilities noted that the owner is provided with appropriate housing and initial food for the rehomed animal(s). For example, the manager of one facility explained how *"The animals are always released with items from their home cages and some diet. We inform the new owners it will take them quite a while to acclimatise and for the initial few days just to place their hands in the cage and let the animals come to them."* Some offered what the NACWO of one facility (which kept a variety of species including horses and beagles) termed a "*going home pack*" which included a bag of the current diet, treats and a vaccination record. One facility mentioned the importance of ensuring the owner is made fully aware of their responsibility for animal wellbeing, and their legal responsibility as a pet owner. As the participant explained, new owners are "*asked to sign a document to confirm they will be responsible for the care and health of the animal(s) and seek veterinary attention should it be required*".

Table 4. Owner preparation frequency count (n = 19).

| Factors to consider as raised by participants | Number of times referenced |
|---|---|
| Rehoming packs | 7 |
| NVS support | 5 |
| Prospective owners invited into facility | 3 |
| Socialisation advice | 2 |
| Legal responsibility | 1 |

This suggests that facilities are aware of the risks in rehoming in terms of liability, and have policy to ensure that owners are legally responsible for their new pet and that transfer of ownership is properly enacted. Three facilities explained that new owners are invited to the facility to view and potentially interact with the animal before rehoming, although again it should be noted that this was mainly for larger companion species such as horses, dogs and cats. In contrast to animal research facilities, which are typically inaccessible to the public, animal shelters tend to welcome visitors, where research finds social interaction to be influential in whether animal are chosen for rehoming [35, 36].

Existing research suggests finding the right sanctuary and environment for the laboratory animal post-rehoming is critical [7, 37]. Further, people who take their newly adopted animal companion to a veterinarian early on are more likely to keep the animal for life [38]—so early and maintained contact with the NVS, which many facilities made an integral part of the rehoming process, is likely to be beneficial.

Another possibility is to rehome the animal with the help of a rehoming organisation (two facilities completing the survey collaborated with external rehoming organisations). This offers advantages; the organisation finds and vets the new owners, provides them with necessary information and remains available as a point of contact. Using such an organisation can be "safe and anonymous" (pg. 2) for the research institution [9]. Indeed, many facilities have already formed good working relationships with such organisations [9]. However, some of those that work at rehoming organisations are volunteers—and literature notes that these volunteers are not always taught the required skills to adequately train and socialise animals [39] In fact, only 12% of animal shelter employees across the United States rated volunteers as well trained "to a great extent" [40]. Research also finds an irregular schedule of social contact with animals from volunteers and frequent changes in active volunteers [41, 42].

From the findings, it is possible to outline the 'typical' 5-stage rehoming process (Fig 2). Participant responses suggested rehoming schemes are catered to the individual animal, and so do divert in some ways from this broad framework. For example, rodents were commonly rehomed with housing, and companion species such as dogs and cats were more likely to be rehomed through third party rehoming organisations, and only after an NVS home visit. However, this diagram broadly conveys the consistent themes that feature in the rehoming process.

## The typical 5-stage rehoming process

**Difficulties encountered by those facilities rehoming.**  The majority (58%) of facilities that had rehomed from 2015–2017 reported that they encountered no problems (Fig 3). However, eight facilities that had rehomed stated that the process was time consuming; one NVS suggested there was a "*delay in Home Office approval*", whilst another facility director suggested there was an extensive "*level of documentation required*". This implies that rehoming is generally considered easy to undertake, but can be time consuming due to the need to navigate complex regulatory boundaries in order to sign the animal off from A(SP)A regulation. Conversely, it is worth noting that very few facilities suggested rehoming was a costly process (2 facilities), that there was difficulty finding homes (1 facility), networking with relevant organisations (1 facility) or that rehoming attracted negative media attention (1 facility).

**Resources needed to rehome.**  As there is an understanding that in order for rehoming to be successful it must be considered carefully and planned thoroughly [43], it is inherently resource intensive for the facility. In the survey, those facilities that had rehomed suggested the main difficulty experienced was that the process was time consuming. Research acknowledges that animals should be spayed prior to rehoming, and that if this is not standard procedure, it can be time-intensive in terms of set up, surgery and aftercare [9]. Working through a third-

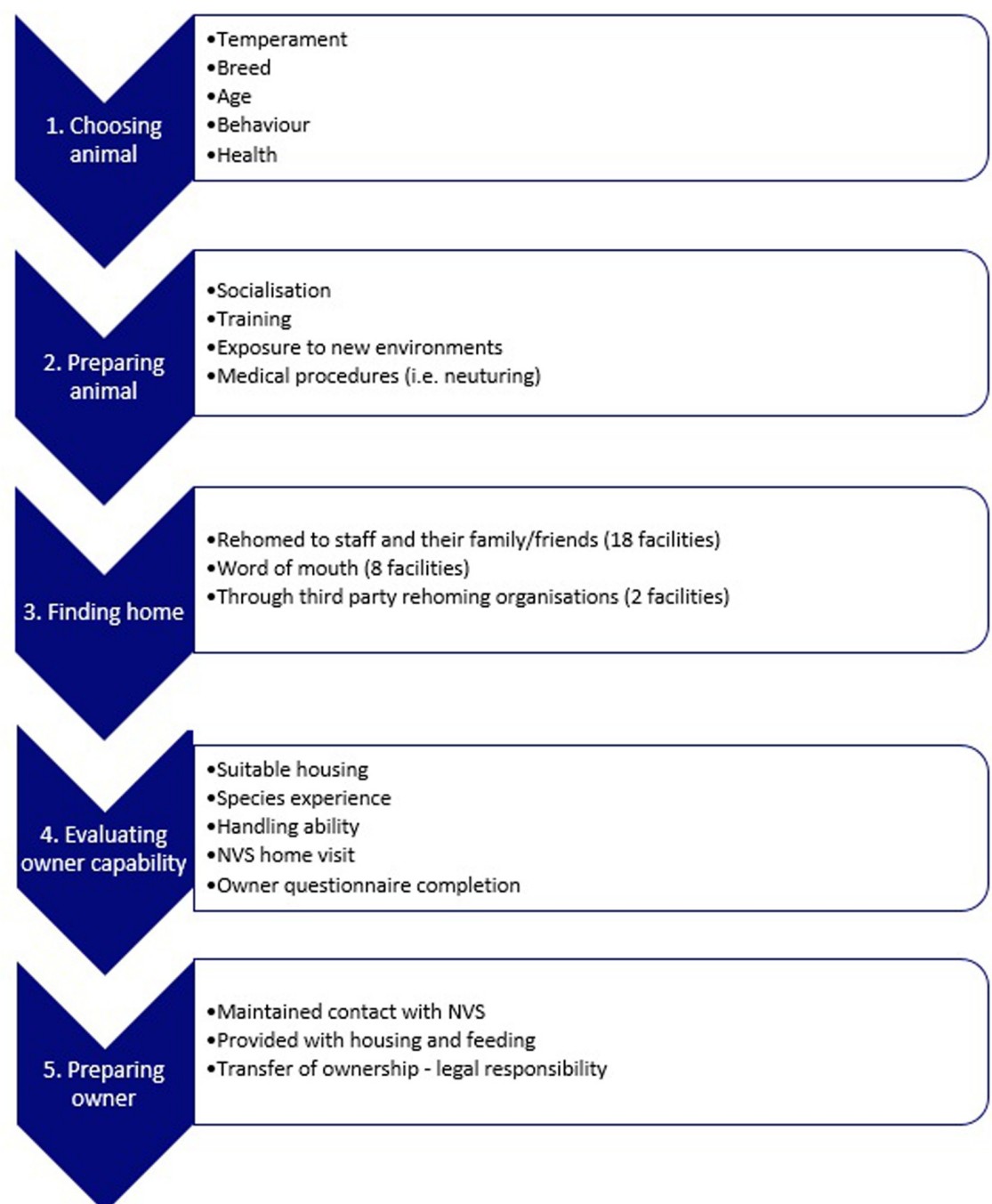

**Fig 2. The 5-stage rehoming process as understood by those UK facilities currently engaging in rehoming.** This flowchart provides an overview of all potential rehoming processes and policies, and not all will be relevant for all research animals.

party rehoming organisation can help to counteract this as the organisation can undertake any medical attention needed, as well as the sourcing and screening of prospective owners. Interestingly, the longer time investment involved in the rehoming of larger animals typically kept as pets (dogs and cats) did not deter efforts to rehome, as these animals were more likely to be considered for rehoming (Table 1). Research recommends that the resources needed to

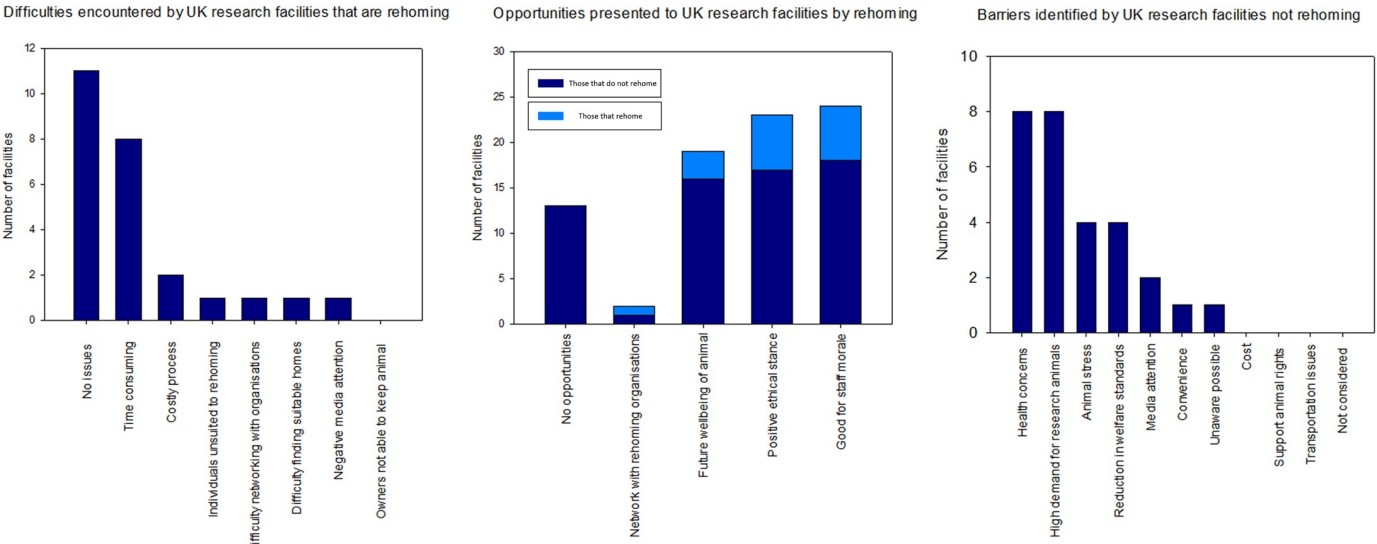

**Fig 3. Graphs which show 1) the main difficulties experienced by UK research facilities that completed the questionnaire that have rehomed in the past 3 years (n = 19), 2) reasons cited by UK research facilities that completed the questionnaire for not engaging in rehoming in the past 3 years (n = 22), and 3) the perceived opportunities presented by rehoming as understood by all UK research facilities that completed the questionnaire (n = 41).**

successfully rehome should not be a deterrent and that it should still be "recommended for the sake of the dogs" (pg. 24) [19].

## Opportunities presented by rehoming

The majority of facilities completing the questionnaire (58%) reflected that rehoming was "good for staff morale" (24 facilities). A similarly high number (23 facilities) believed it showed a positive ethical stance. The expectation of future well-being of animals played a significant, but slightly lesser role (19 facilities), while 13 facilities felt rehoming offered no opportunities (Fig 3). Interestingly, but perhaps unsurprisingly, those facilities not currently engaging in rehoming did not reflect that the process could benefit them. Conversely, those that had engaged in rehoming in the previous 3 years were more likely to suggest it contributed positively to animal welfare, staff morale and demonstrated a positive ethical stance.

Existing research reports that rehoming can benefit animal welfare, allowing a dignified and deserved retirement [44]. Rehoming helps to 'uphold scientist's ethical responsibilities' [18]. Rehoming also has important ramifications for the wider facility and can help to develop and foster a "culture of care" and staff wellbeing [7, 43], as echoed in the questionnaire results (Fig 3). Staff morale may be improved further if the staff member is able to rehome the animals themselves, which the survey reflected was fairly common; 18 facilities rehomed their animals to staff members. Routine killing is emotionally challenging and stressful for facility staff [45], so any opportunity to allow animals to have a life outside of the laboratory will benefit employees, most of whom care deeply about the animals with which they work [46]. There thus exists a two-way process, whereby it is positive for staff to 1) circumvent the emotional stress of unnecessary killing (avoidance of negative states), and 2) provide the animal with an increased quality of life once rehomed (promotion of positive states). However, there are challenges that come with rehoming to staff—an NVS explained how the rodents could not be rehomed to employees as staff might "*acquire rodents from other sources*" that are "*microbiologically dirty [. . .] which could present a risk of inadvertent delivery of disease*". Microbiological

contamination is thus considered a risk, and therefore some facility staff cannot keep rodents as pets, including those from the laboratory.

### Reasons facilities are not currently engaged in rehoming

Amongst those facilities that had not rehomed in the previous 3 years, eight reported that the reason was concern for the animal's health if it were to be rehomed. Eight stated that high demand means few are left to retire. Slightly fewer numbers felt rehoming would be too stressful for the animal (4 facilities), that it was difficult to predict long-term health implications (4 facilities) or that rehoming would result in a loss of control (4 facilities). Fear of unwanted or negative media attention (2 facilities), convenience (one facility) and being unaware that rehoming is possible (one facility) were rarely selected as reasons not to rehome (Fig 3). Reasons for not rehoming can thus be grouped into welfare concerns with regard to the animal's health if rehomed, practical issues surrounding demand and the fact that research needs' tend to leave few animals to retire, and external challenges including fear of negative media attention.

As well as the challenges recognised above (Fig 3), utilising the open answer box, participants articulated additional reasons for their lack of engagement in rehoming. These included that; 1) rehoming is impossible for some GA animals due to its illegal nature following A(SP)A policy [1] (if GA animals are rehomed, the environment and other animals may be exposed to infectious agents and consequently rehoming could have severe negative impacts that extend beyond the boundaries of the laboratory), 2) three facilities explained that they had never been approached to rehome, and therefore a lack of demand for rehoming was cited as a barrier, and finally 3) animals were not considered for rehoming at some facilities because of the research undertaken, much of which was of a terminal nature due to the tissues required. As one participant, an NACWO at an amphibian and rodent facility, explained, the "*vast majority of projects involve terminal or non-recovery final procedures.*" The survey responses thus reflect a conflict between the demands of the research and any possibility of rehoming.

**Biosecurity.** Despite it being scientifically unclear exactly how some GA animals might pose a biosecurity risk, the rehoming of GA animals was considered illegal by some who responded to the survey. One participant, a Director at a rodent facility, explained how their genetically altered rodents are "not permitted for rehoming". Many UK facilities house genetically modified animals, and this number is increasing [24–26]. The Home Office advice note on rehoming states that an animal should only be rehomed if it will not harm the environment, other animals, itself or people [1], but participants acknowledged the difficulties in guaranteeing this when rehoming genetically modified laboratory animals. As one participant, an AWERB Chair at a fish and rodent facility explained, "We are working almost exclusively with infectious pathogens so rehoming cannot be achieved from a human safety point of view". Another, the AWERB chair at a rodent, fish and pig facility, discussed the "associated risks" involved with rehoming GA animals used in "infectious work". Thus, health risks from exposure to animals outside of the laboratory limits rehoming practice. A case study of GA pig rehoming discusses how 'EnviroPigs' were not rehomed for fear of potential environmental and food safety risks, making transfer to a farm sanctuary irresponsible [27]. Indeed, there is uncertainty around allowing transgenic animals to be retired [9], as well as complex legal liability issues should animals escape [27]. Research advises that no genetically modified livestock be rehomed and enter the food chain [9]. It is also suggested that genetically modified animals should not be adopted, importantly whether they are neutered or intact, to any member of the public [9].

**Perceptions of risk.**   Linking to this are issues of liability, risk and reputation for the research facility. Participants noted the importance of the legal transfer of ownership when rehoming—implying there exists a perceived risk for the facility otherwise. One participant, a NACWO, explained in response to an open answer in the questionnaire survey, "*The new owner is made fully aware of their responsibility for the pony's health and well-being. They are also made aware of their legal responsibility to change ownership details on the passport*". Ultimately, facilities cannot control how the rehoming process may reflect on them and their reputation [9]. This is also reflected in two facilities citing negative media attention as a barrier to rehoming (Fig 3). Research finds a major deterrent to rehoming is that sanctuaries/members of the public could discuss abnormal animal behaviour/physiology with the media and thus the facility be presented in a negative light [47]. This could result in them being unlikely to rehome again. To combat this, there is potential to introduce confidentiality agreements regarding the origins of the animal before considering rehoming [47].

**Animal welfare concerns.**   Although also acting as a motivation to rehome, perceived future animal welfare also represented a barrier in terms of loss of control and potential reduced standards of care. There were worries regarding ensuring animal welfare once the animal had left the facility—one participant, a NACWO at a fish and rodent facility, explained how it was "*too difficult to monitor animal welfare after re-homing has occurred*". There were also more specific concerns over the conditions in which animals will be kept; one participant, a manager at a mouse facility, outlined that the animals at their facility required "*high standards of care not readily available*". Worries regarding the animals' life post-rehoming are justified; there exists a high turnover rate of companion animals, and owners may lose interest and get rid of the animal [9]. Previous research has also outlined worries regarding rehoming psychologically distressed monkeys, who may not adapt well to a new environment [7]. The transportation of animals, especially primates who possess significant mental capacities [48], can also be very stressful [49]. Ultimately, participants eluded that the ethical legislation that exists in facilities does not extend to private homes/sanctuaries, and there were hence worries regarding ensuring animals were provided with a life worth living post-rehoming.

## Study limitations

It should be noted that the collected responses may represent a selection bias, as those who rehome more regularly may also have been more likely to complete the survey. As it is a controversial issue, there may have been a social desirability bias in response to more subjective questions. Additionally, although over 25% of facilities in the UK completed the questionnaire, this remains a sub sample of the total population. Thus, caution should be exercised when attempting to generalise results to all UK research facilities. Finally, an extrapolation was made from 2017 Home Office data to the years 2015 and 2016 to calculate an approximate number of surplus animals across the 3 years. However, there is likely to be some variance in the proportions of surplus animals by species year by year that is not accounted for in this research.

## Future research

Although this research demonstrates that some species are preferred over others for rehoming, future research should seek to understand the mechanisms by which this occurs. This is necessary to understand why particular animals are chosen for rehoming, and may promote the rehoming of animals not typically considered. Work should also be undertaken to evaluate the success of rehoming schemes with species including rodents and fish, not simply larger animals traditionally kept as pets or primates as has been undertaken previously. Despite the clear impossibility of finding homes for all of the rodents and fish currently kept in UK facilities,

there would be merit in understanding the drivers and processes of rehoming even small numbers of these species. This would help to encourage and support facilities with species such as rats and mice in their efforts to rehome, as well as guiding them in developing relevant and useful facility-wide policy.

## Conclusion

This research has demonstrated that rehoming occurs in just under 50% of the UK research facilities that participated in the study, but is usually in very small numbers (just 2322 animals are known to have been rehomed from 2015–2017). There exists a clear species preference for rehoming, whereby traditional companion animals (cats, dogs and horses) are more commonly considered. Rehoming appears to occur through two pathways: 1) in small numbers of rodents (typically gerbils, rats, guinea pigs and rabbits) rehomed to staff and their families and friends, and 2) in larger numbers of traditional companion animals through extensive public rehoming schemes. The main motivation for doing so is to boost staff morale and demonstrate a positive ethical profile. Expectation of future well-being of the animal also played a slightly lesser, but still noteworthy and connected role. These benefits were not realised by those facilities not engaging in rehoming.

The most significant barrier is the time taken to rehome, yet generally most facilities that rehomed in the previous 3 years found the process to be easy and few experienced substantial difficulties. This may be because the survey revealed that rehoming is generally a very well-planned process, with 14 out of the 19 facilities that had rehomed in the 2015–2017 period employing facility-wide rehoming policy which included choosing the appropriate animals, socialisation and training, and owner selection and preparation. This importantly differed through its tailoring to the animal in question.

The main reasons for choosing not to rehome include concern for the animal's health if it were to be rehomed, high demand for research animals, and animals in the facility being unsuitable for rehoming (participants explained this was primarily genetically modified fish and mice).

The questionnaire revealed that rehoming is a known and considered pathway for laboratory animals, but is undertaken in relatively small numbers. Despite rehoming occurring in low numbers, the practice was interpreted by facilities that rehomed to be good for animal welfare, the staff and the wider facility, and as such should be considered where possible.

As part of a movement to enhance animal welfare and introduce measures which promote a 'life worth living' through the attainment of positive states in animals, rehoming from UK animal research facilities finds a place. Rehoming helps to support staff in overcoming issues related to moral stress, as well as address public concern regarding the current routine nature of animal killing in the laboratory. In order to promote the practice, it is imperative to enhance current understandings regarding both which facilities are participating in the practice, and how they do it, in order to disseminate the information to institutions not currently active in this area.

## Supporting information

**S1 Appendix.**
(DOCX)

## Acknowledgments

We thank all participants for their generous time contribution, and to the reviewers for their valuable comments.

## Author Contributions

**Conceptualization:** Tess Skidmore.

**Data curation:** Tess Skidmore.

**Formal analysis:** Tess Skidmore.

**Funding acquisition:** Emma Roe.

**Investigation:** Tess Skidmore.

**Methodology:** Tess Skidmore.

**Supervision:** Emma Roe.

**Writing – original draft:** Tess Skidmore.

**Writing – review & editing:** Tess Skidmore, Emma Roe.

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
