## [Editor Report · Decision Letter 0]

23 Jan 2020

PONE-D-20-01484

A quantitative investigation into the occurrence and processes of laboratory animal rehoming from UK animal research facilities

PLOS ONE

Dear Ms Skidmore,

Before this paper can be sent out for review, it needs to be condensed. In its present form, the introduction and the discussion are 7-8 pages each. This is not justified by the complexity of the study, which is a questionnaire study of a mainly descriptive nature. Please see how you can shorten and condense.

Please note that this request for revision is sent before the peer review process has started. The final decision on the submission will depend on the outcome of the peer review.

We would appreciate receiving your revised manuscript by Mar 08 2020 11:59PM. To enhance the reproducibility of your results, we recommend that if applicable you deposit your laboratory protocols in protocols.io, where a protocol can be assigned its own identifier (DOI) such that it can be cited independently in the future. For instructions see: http://journals.plos.org/plosone/s/submission-guidelines#loc-laboratory-protocols

We look forward to receiving your revised manuscript.

Kind regards,

I Anna S Olsson, Ph.D.

Academic Editor

PLOS ONE

Journal Requirements:

2. Please consider changing the title so as to meet our title format requirement (https://journals.plos.org/plosone/s/submission-guidelines). In particular, the title should be "Specific, descriptive, concise, and comprehensible to readers outside the field" and in this case it is not informative and specific about your study's scope and methodology. In particular, as you mention in your text, it is not clear that a survey-based study with this response rate allows you to generalize to all animal facilities, and therefore it's not clear that your study is a quantitative analysis of all facilities; moreover, your study also includes a qualitative component.

3. In your figures (in particular fig. 1), please clarify that only facilities from which you obtained responses are included.

4. Please do not include funding sources in the Acknowledgments or anywhere else in the manuscript file. Funding information should only be entered in the financial disclosure section of the submission system. https://journals.plos.org/plosone/s/submission-guidelines#loc-acknowledgments.

5. Please include additional information regarding the survey or questionnaire used in the study and ensure that you have provided sufficient details that others could replicate the analyses. For instance, if you developed a questionnaire as part of this study and it is not under a copyright more restrictive than CC-BY, please include a copy, in both the original language and English, as Supporting Information.

6. We note that you have indicated that data from this study are available upon request. PLOS only allows data to be available upon request if there are legal or ethical restrictions on sharing data publicly. For more information on unacceptable data access restrictions, please see http://journals.plos.org/plosone/s/data-availability#loc-unacceptable-data-access-restrictions.
---

## [Author Response · Author response to Decision Letter 0]

17 Feb 2020

Additional revision:

1. Thank you for useful considerations regarding data restrictions. Unfortunately, I cannot share the data for ethical reasons as consent was on the basis that the information gathered from participants would not be shared beyond the authors of this paper. I proceeded in this way as it is difficult to fully anonymise data in a small field. I hope this provides a satisfactory explanation. Thank you again for your comments.

---

## [Decision Letter · Decision Letter 1]

14 Apr 2020

PONE-D-20-01484R1

A semi-structured questionnaire survey of laboratory animal rehoming practice across 41 UK animal research facilities

PLOS ONE

Dear Ms Skidmore,

Thank you for submitting your manuscript to PLOS ONE. After careful consideration, we feel that it has merit but does not fully meet PLOS ONE’s publication criteria as it currently stands. Therefore, we invite you to submit a revised version of the manuscript that addresses the points raised during the review process.

The two reviewers are both of the opinion that the topic is relevant and that there is merit in the work, but that the manuscript needs substantial revision. Please see the detailed review reports in the following for issues that need to be addressed in the manuscript. 

We would appreciate receiving your revised manuscript by May 29 2020 11:59PM. To enhance the reproducibility of your results, we recommend that if applicable you deposit your laboratory protocols in protocols.io, where a protocol can be assigned its own identifier (DOI) such that it can be cited independently in the future. For instructions see: http://journals.plos.org/plosone/s/submission-guidelines#loc-laboratory-protocols

We look forward to receiving your revised manuscript.

Kind regards,

I Anna S Olsson, Ph.D.

Academic Editor

PLOS ONE

Reviewers' comments:

Reviewer's Responses to Questions

**Comments to the Author**

1. If the authors have adequately addressed your comments raised in a previous round of review and you feel that this manuscript is now acceptable for publication, you may indicate that here to bypass the “Comments to the Author” section, enter your conflict of interest statement in the “Confidential to Editor” section, and submit your "Accept" recommendation.

Reviewer #1: (No Response)

Reviewer #2: (No Response)

2. Is the manuscript technically sound, and do the data support the conclusions?

Reviewer #1: Partly

Reviewer #2: No

3. Has the statistical analysis been performed appropriately and rigorously? 

Reviewer #1: No

Reviewer #2: N/A

4. Have the authors made all data underlying the findings in their manuscript fully available?

Reviewer #1: No

Reviewer #2: No

5. Is the manuscript presented in an intelligible fashion and written in standard English?

Reviewer #1: Yes

Reviewer #2: No

6. Review Comments to the Author

Reviewer #1: The study has an important topic: The authors carried out a survey on the rehoming practice of 41 research facilities in the UK to find answers for the following questions: How often are animals rehomed, what attitudes exist in the facilities about this topic and which reasons prevent institutions from rehoming their laboratory animals.

I propose a major revision because the authors should present their results in a more comprehensive manner.

• Please present all results of your questionnaire in the manuscript. I miss specific data, for example: How many animals of which species are kept in the 41 facilities that answered? How many animals of which species they rehomed trough which pathway (staff, third party organization)? How often have animals been given to schools, farms etc.? In the chapter “The rehoming process” (lines 349-419) I miss specific data to follow up on the statements. Example: Line 359: “The majority of respondents…” - what number exactly?

The presentation of results is more comprehensible in the chapters in lines 431-504 (chapters “Difficulties encountered…” to ”Reasons facilities are not… ”).

• In my opinion it is misleading to present the number of the rehomed animals as a percentage of all animals kept in the UK. Please place the number of the rehomed animals in relation to the animals kept in the 41 facilities that responded.

• In the method chapter (for example lines 239-241) and in the results section the authors already included discussion aspects. Please separate these areas. PLOS One allows a combination of results and discussion, but this should be shown in the heading accordingly.

Further comments:

• Please include more data in the abstract.

• Please include the legal requirements in the UK into the introduction. Directive 2010/63/EU makes specific statements on what to do with animals at the end of procedures and on the rehoming of animals.

• Please explain abbreviations when you use them for the first time (e.g. GA in line 216)

• You can delete the sentence in lines 339-340 “The next section…”

• Please add the n-numbers to the figure captions for figures 7 and 8 (n = 22 facilities without rehoming…)

• I do not understand why the figures are referred to here: line 374 for figure 4 and line 541 for figure 8?

• Figure 1 is unnecessary as this information is expressed in figure 2.

• Table 2 contains very little information and is unnecessary since you can also insert the missing numbers in the text.

• How did you calculate the numbers in figure 4? Birds 36.65%?

• Line 344 and line 656: Rabbits and ferrets are not rodents! Replace with “small mammals”.

• In the illustrations, the font is sometimes too small and difficult to read, e.g. in figures 6 and 7.

• Discussion lines 644 ff. Do you have an idea how to find a new home for millions of fish, rats and mice? Is that realistic? Please discuss.

Reviewer #2: This paper aims to assess how many facilities in the UK rehome former research animals, which species, which numbers, for which reasons and by which process. The authors also look at barriers to rehoming. In itself, this is a very relevant and interesting theme to explore, as there are insufficient data to support the practical implementation of the rehoming stipulations in legislation.

However, I have some major concerns with this paper In summary, some parts of the paper lack focus (introduction, discussion), there is an issue with confusing the concepts of animal welfare and ethics in discussing arguments for rehoming, I have doubts about the way the numerical data are presented (known percentage of rehomed animals), the paper lacks critical use of references (there are references to books that are implicitly referred to as results from scientific research, which they aren’t) and the text as a whole does not read fluently. I lost track of the flow several times and had to go back and re-read.

I will begin with providing detailed comments on my major concerns.

ABSTRACT

The data about rehomed versus kept are misleading. See comment regarding major concern about the analysis.

INTRODUCTION

- I do not agree with the authors approach to refute "death of an animal does not constitute a welfare matter". The authors seem to confuse the concepts of animal welfare and animal ethics. Animal Welfare concerns the situation of the animal as the animal perceives it. In that respect, the fact that an animal dies is not an animal welfare matter, as we have no indication that animals are conscious of any future time that is lost by their death. HOW the dies (i.e. how much suffering does the concept induce) is an animal welfare matter. Animal ethics, on the contrary, is how we humans think animals should be treated. In that respect, giving an animal " a life worth living" after retirement from the research facility is an ethical matter, in which animal welfare aspects have to be taken into account. The authors should rethink their line of reasoning.

- The introduction lacks focus and is unbalanced, being very one-sided towards an obligation to rehome. This should be more balanced for a scientific paper in my view. I would propose to structure the introduction to make it very concise and clean, including: what is rehoming, what are animal welfare benefits and risks, what are ethical arguments for or against rehoming (if that is indeed something they will get back to based on the survey responses; if not, I think the authors should consider to stay away from philosophical/moral discussions).

METHODOLOGY

My one issue with the methodology is how the authors choose to calculate the known percentage of rehomed animals. The current way to calculate this (numbers rehomed across 41 facilities compared to all UK unused non-GA + first time procedure animals) provides a huge underestimation. The authors do state twice in the paper that they calculate the “minimum percentage” and yet, in the rest of the manuscript, they treat it as if it is the maximum. A more correct calculation would have been to use numbers of rehomed animals in relation to number of kept animals for the 41 facilities. As I look at the questionnaire, however, I see that the authors only have information about “animals kept at this time”, for which I don’t know which period this is. The only somewhat correct way to present the data then, would a comparison between species, using the proportion of each species that is being rehomed (based on 41 facilities) and proportion of each species as they are used throughout the UK, as an indication of the disproportion, but without comparing numbers kept to numbers rehomed.

RESULTS

Following my major concern in the methodology, the results section needs to be re-worked.

DISCUSSION

The discussion does not read easily, I think in part because a clear structure is generally lacking. It is difficult to see whether the authors use the same order as the one that was used to present the results, or whether they refer back to the research questions and deal with them in sequence. Perhaps a few subtitles could help.

Some elements of the discussion do not appear in the results section. For example Lines 532-533: the view of the respondents about which species are ‘unrehomeable’ was not asked directly of the particpants. This may have come out in the open questions, but then the results from a structured analysis on this must be presented in the results section. Another one is on line 546: biosecurity. This is the only time in the entire manuscript that biosecurity is mentioned as a significant barrier. This must be presented in the results section first.

I would not use single quotes to support statements if you have not presented a structured analysis of open questions in the results section. This makes the study look extremely subjective and anecdotal.

CONCLUSION

The conclusion is actually quite concise and clear. But its contents needs to be modified to reflect the requested changes.

Next, there are also some minor concerns.

ABSTRACT

Lines 30-32: the sentence "this paper reports on research findings that explain..." is too strong. The findings do not explain anything, they merely provide an indication.

Line 36 and beyond: the abstract should at least mention the number of facilities that participated and the response rate.

Line 36-37: This needs to be adapted to respond to my major comment in methodology section about the analysis

INTRODUCTION

Line 58: “The practice supports the notion that…”. I do not agree with “supports”. The practice of rehoming is guided by the notion that animals are sentient beings etc.

Lines 69-70: these arguments are worded too strong. They may be possible, but certainly not true for all cases. Please re-word.

Line 78-79: again too strong. In veterinary practice, euthanasia is also frowned upon many times. And routine euthanasia (which can happen e.g. in an overcrowded shelter) is never regarded as favourable.

Line 91 and beyond: I am concerned about the authors’ understanding of the concept of animal welfare, because of the references they use in this section. The fact that they refer to a philosophical paper to define animal welfare concerns me. There are good papers by animal welfare scientists, which should be preferred as a source. See for example: Broom 2011, Acta Biother, 59:121-137 or Weary and Robbins 2019, Anim Welf, 28:33-40.

Line 130-142: I find this paragraph confusing. The authors could elaborate a bit on the results from the outcome studies, to indicate good practices in rehoming already.

Line 133: “The overwhelming focus on these species is limiting”: I do not understand what the authors are trying to say.

Line 134-137: The references 5 and 20 are books. Very unlikely that these report original research. Can primary sources be used? Or are the authors of the books referring to anecdotal evidence? This needs to be made more clear, because the authors state “as research points out…”, so the supporting references should be research papers.

Line 148: “This is necessary to enhance animal welfare”: I find this quite a vague statement and would like to know what the authors mean. Is it about enhancing animal welfare by rehoming, or enhancing animal welfare by rehoming correctly (using proper procedures, which can help guarantee the welfare of the animals after rehoming, as demanded by law)?

METHODOLOGY

The methods section needs to mention when the survey was administered (months & year).

Line 178 and 242: “the university of XXXXX” – I assume the authors have blinded the university, but since this is not a blinded review, that is not necessary.

Line 179: “spilt” should be “split”

Line 192: AWERB should appear in full the first time it is used

Lines 192-195: this information should be presented at the beginning of the results section

Line 216 and 496 (for example): sometimes GA is used and sometimes GM, while I think the authors mean the same

Line 218: No need for this long sentence, just use the term that was introduced before: surplus animals

Lines 207-235: The data processing section is too long and fragmented. Also, crucial information is missing. Please create one or two paragraphs with data processing information, concisely reporting how the responses to the questions were processed (how were closed questions processed, what about open questions – was thematic analysis used?), which variables were calculated, why and how….

RESULTS

Lines 259-261: this is a bold statement that is insufficiently supported by the data that are presented. How does the variety in the survey relate to the variety in all other facilities?

Figure 1: I would delete figure 1. There is no added value there, as the numbers are already in the text.

Line 278-280: I find this an odd statement, given that, equally, it is very possible institutions that do not rehome did not complete the survey. Also, does this mean that incomplete surveys were not counted? This needs to be stated in the data processing part of the Methodology section.

Line 289: “If we assume that the ratio above extends” – to what?

Line 291: This is an example of how minimum percentage is not used throughout. Because it is a minimum, you can only state that “at least 0.02% of laboratory animals are known to be rehomed from 2015-2017”, but again, this figure to me is really misleading as it is a huge underestimation.

Line 300: keep in mind that primates may not be rehomed in the UK sense of the word, but they may be retired from research, while still living at the facility.

Table 1: again, misleading due to the “known percentage rehomed” calculation. Similar comment for the last column with the ration.

Figure 4: The left pie chart is difficult to read. I would turn this figure into a table comparing proportions of animals rehomed (41 institutions) and animals kept (UK total). This, in my opinion is the only valid numerical comparison that could be made (see my major concern about the calculation of “known percentage rehomed”).

Lines 349 and beyond: the presentation of results is very qualitative as it almost looks at individual responses from respondents. I feel a more structured analysis, like a thematic analysis with a frequency count, is appropriate.

Lines 487-489: The grouping of reasons for not rehoming does not adequately capture the responses. E.g. Fear of unwanted or negative media attention is not a practical issue. Please re-consider this.

DISCUSSION

Lines 526-527: “this is also demonstrated by only one facility of the 41 facilities”: it is not clear what is demonstrated

Line 545 and beyond: the authors seem to suggest that all GM models pose a biosecurity risk, which is not true.

Line 585: “this effect” – which effect? Please state clearly to what you are referring.

7. PLOS authors have the option to publish the peer review history of their article (what does this mean?). If published, this will include your full peer review and any attached files.

Reviewer #1: No

Reviewer #2: No

---

## [Author Response · Author response to Decision Letter 1]

7 May 2020

Thank you for your very helpful comments on the first draft of this paper. I have responded to them below. 

Reviewer 1:

• Please present all results of your questionnaire in the manuscript. I miss specific data, for example: How many animals of which species are kept in the 41 facilities that answered? How many animals of which species they rehomed trough which pathway (staff, third party organization)? How often have animals been given to schools, farms etc.? Thank you for this contribution. Unfortunately, however, we cannot provide this information from the data collected. The data we have does not break down the different pathways used to rehome to an individual/species level. We only have data for the numbers of facilities that use different pathways to rehome. In the chapter “The rehoming process” (lines 349-419) I miss specific data to follow up on the statements. Example: Line 359: “The majority of respondents…” - what number exactly? This has now been changed in accordance with comments – we have added in the specific number of respondents. On reviewer 2’s suggestion, this has also been supplemented with a frequency count of particular themes arising from the qualitative open answer responses. 

The presentation of results is more comprehensible in the chapters in lines 431-504 (chapters “Difficulties encountered…” to ”Reasons facilities are not… ”). Thank you, this has been noted. 

• In my opinion it is misleading to present the number of the rehomed animals as a percentage of all animals kept in the UK. Please place the number of the rehomed animals in relation to the animals kept in the 41 facilities that responded. Unfortunately, we cannot place the number of animals rehomed in relation to the number of animals kept across the 41 facilities across the 3 years because the data we have is only for animals kept in the 41 facilities at that time. Thus, and in keeping with reviewer 2’s comments, we have removed the suggestion that the number of rehomed animals is a percentage of all animals kept, and instead presented the different data to allow a visual comparison (see tables 1 and 2). 

• In the method chapter (for example lines 239-241) and in the results section the authors already included discussion aspects. Please separate these areas. PLOS One allows a combination of results and discussion, but this should be shown in the heading accordingly. Thank you for this advice, the results and the discussion sections have now been combined, and the headings adjusted accordingly. 

Further comments:

• Please include more data in the abstract. We have now added in information regarding number of facilities that completed the survey and the response rate, number of animals known to be rehomed (2322 animals), and a species breakdown of numbers rehomed: “although 94.15% of species kept in laboratories are rodents, they make up under a fifth (19.14%) of all animals known to be rehomed between 2015-2017.”

• Please include the legal requirements in the UK into the introduction. Directive 2010/63/EU makes specific statements on what to do with animals at the end of procedures and on the rehoming of animals. Thank you, this is a very useful suggestion. The legal requirements regarding rehoming as laid out in Directive 2010/63/EU have now been added to the introduction.

• Please explain abbreviations when you use them for the first time (e.g. GA in line 216). Now corrected this by explaining the abbreviation. 

• You can delete the sentence in lines 339-340 “The next section…” Thank you, this line has now been deleted. 

• Please add the n-numbers to the figure captions for figures 7 and 8 (n = 22 facilities without rehoming…) We have updated figure captions to reflect the number included in the figure.

• I do not understand why the figures are referred to here: line 374 for figure 4 and line 541 for figure 8? The reference to these figures have now been removed. 

• Figure 1 is unnecessary as this information is expressed in figure 2. Thank you. Figure 1 has been removed in line with the recommendation of both reviewers 1 and 2 

• Table 2 contains very little information and is unnecessary since you can also insert the missing numbers in the text. Now removed in line with reviewer’s recommendation. 

• How did you calculate the numbers in figure 4? Birds 36.65%? A more detailed description of how we calculated the numbers has now been included in the figure caption for figure 4. It reads as follows “The left column shows the percentage of animal groups (such as small mammals, birds, agricultural animals) currently kept in all UK facilities. This data is based on Home Office annual statistics of animals used in research. The right column shows an equal breakdown of animal groups rehomed across the 41 facilities that completed the survey to allow a comparison”. 

• Line 344 and line 656: Rabbits and ferrets are not rodents! Replace with “small mammals”. Thank you – this has now been changed to small mammals. 

• In the illustrations, the font is sometimes too small and difficult to read, e.g. in figures 6 and 7. The font size has now been made larger in these figures to ensure it is more readable. 

• Discussion lines 644 ff. Do you have an idea how to find a new home for millions of fish, rats and mice? Is that realistic? Please discuss. Thank you for this consideration. We feel it is not realistic to try to rehome all of these species. However, it is important that, because of this, research does not neglect these species. Instead, research should be undertaken into the rehoming of these species, as well as the current focus on large mammals (dogs, cats, primates). We have now added in “This does not mean rehoming all those fish and rodents currently kept in UK facilities, as given the numbers this would be unrealistic, but instead understanding why and how efforts are made to rehome even small numbers of these species”. 

Reviewer 2:

This paper aims to assess how many facilities in the UK rehome former research animals, which species, which numbers, for which reasons and by which process. The authors also look at barriers to rehoming. In itself, this is a very relevant and interesting theme to explore, as there are insufficient data to support the practical implementation of the rehoming stipulations in legislation.

However, I have some major concerns with this paper In summary, some parts of the paper lack focus (introduction, discussion), there is an issue with confusing the concepts of animal welfare and ethics in discussing arguments for rehoming, I have doubts about the way the numerical data are presented (known percentage of rehomed animals), the paper lacks critical use of references (there are references to books that are implicitly referred to as results from scientific research, which they aren’t) and the text as a whole does not read fluently. I lost track of the flow several times and had to go back and re-read.

I will begin with providing detailed comments on my major concerns.

ABSTRACT

The data about rehomed versus kept are misleading. See comment regarding major concern about the analysis. This has now been updated and does not discuss data about rehomed v. kept. 

INTRODUCTION

- I do not agree with the authors approach to refute "death of an animal does not constitute a welfare matter". The authors seem to confuse the concepts of animal welfare and animal ethics. Animal Welfare concerns the situation of the animal as the animal perceives it. In that respect, the fact that an animal dies is not an animal welfare matter, as we have no indication that animals are conscious of any future time that is lost by their death. HOW the dies (i.e. how much suffering does the concept induce) is an animal welfare matter. Animal ethics, on the contrary, is how we humans think animals should be treated. In that respect, giving an animal " a life worth living" after retirement from the research facility is an ethical matter, in which animal welfare aspects have to be taken into account. The authors should rethink their line of reasoning. The discussion of death not constituting a welfare issue has been removed, and instead effort has been made to explain how rehoming may both improve animal welfare when following regulatory guidance, but also potentially also compromise it. 

- The introduction lacks focus and is unbalanced, being very one-sided towards an obligation to rehome. This should be more balanced for a scientific paper in my view. I would propose to structure the introduction to make it very concise and clean, including: what is rehoming, what are animal welfare benefits and risks, what are ethical arguments for or against rehoming (if that is indeed something they will get back to based on the survey responses; if not, I think the authors should consider to stay away from philosophical/moral discussions).Thank you for this useful suggestion. The moral and ethical arguments for considering rehoming have now been removed. In order to balance out our arguments, we have added a discussion of how rehoming can also negatively impact welfare: “Of course, it is also crucial to acknowledge that rehoming is not always in the best interests of the animal, and may instead only serve to compromise welfare. Current UK regulatory guidance, such as the Animals in Scientific Procedures Act, or A(SP)A, helps to ensure welfare is maintained in the laboratory, yet once rehomed this legislation is no longer in place to protect the animal in question. It is thus the facility’s responsibility to ensure that rehoming will in no way compromise welfare [22]. This must remain an absolute priority. Factors that can affect welfare during the rehoming process are the animal’s state of health, duration and condition of transport to new home, and the social and/or physical environment the animal will be moved to [8]. Research notes that potential compromises to welfare can be minimised or eradicated with careful and thorough planning of key processes such as transport [8] and socialisation [10, 22] to lower animal stress.”

METHODOLOGY

My one issue with the methodology is how the authors choose to calculate the known percentage of rehomed animals. The current way to calculate this (numbers rehomed across 41 facilities compared to all UK unused non-GA + first time procedure animals) provides a huge underestimation. The authors do state twice in the paper that they calculate the “minimum percentage” and yet, in the rest of the manuscript, they treat it as if it is the maximum. A more correct calculation would have been to use numbers of rehomed animals in relation to number of kept animals for the 41 facilities. As I look at the questionnaire, however, I see that the authors only have information about “animals kept at this time”, for which I don’t know which period this is. The only somewhat correct way to present the data then, would a comparison between species, using the proportion of each species that is being rehomed (based on 41 facilities) and proportion of each species as they are used throughout the UK, as an indication of the disproportion, but without comparing numbers kept to numbers rehomed. Calculating the percentage of animals rehomed as a percentage of Home Office figures has now been removed, and instead a visual comparison has been enabled through tables 1 and 2, which show the number of animals kept (in all UK research facilities between 2015-2017) and those rehomed (out of the 41 facilities that completed the questionnaire). 

RESULTS

Following my major concern in the methodology, the results section needs to be re-worked. Thank you – the results section has now been updated to reflect changes in the analysis. 

DISCUSSION

The discussion does not read easily, I think in part because a clear structure is generally lacking. It is difficult to see whether the authors use the same order as the one that was used to present the results, or whether they refer back to the research questions and deal with them in sequence. Perhaps a few subtitles could help. Thanks, this is a very useful suggestion. The results and conclusions sections have now been merged, and extra subtitles have been added to aid with a clearer structure.

Some elements of the discussion do not appear in the results section. For example Lines 532-533: the view of the respondents about which species are ‘unrehomeable’ was not asked directly of the participants. This may have come out in the open questions, but then the results from a structured analysis on this must be presented in the results section. Another one is on line 546: biosecurity. This is the only time in the entire manuscript that biosecurity is mentioned as a significant barrier. This must be presented in the results section first. Thank you for this useful advice. A structured analysis has now been provided in the results and discussion section, and it has been made clearer that biosecurity risks as a barrier have come from the open question responses. 

I would not use single quotes to support statements if you have not presented a structured analysis of open questions in the results section. This makes the study look extremely subjective and anecdotal. We have now provided a structured analysis of the open questions, including a frequency count. We have consequently left in the single quotes to support statements, as the structured analysis will make their use more objective and scientific. 

CONCLUSION

The conclusion is actually quite concise and clear. But its contents needs to be modified to reflect the requested changes. Thank you for your positive comments. The conclusion has now been updated in line with wider changes to the paper. 

Next, there are also some minor concerns.

ABSTRACT

Lines 30-32: the sentence "this paper reports on research findings that explain..." is too strong. The findings do not explain anything, they merely provide an indication. I have adapted this as suggested to: “This paper reports on research findings that provide an indication of the uptake of animal rehoming by UK facilities...”

Line 36 and beyond: the abstract should at least mention the number of facilities that participated and the response rate. Thank you. These figures have now been included in the abstract, along with more figures in line with reviewer 1 comments. 

Line 36-37: This needs to be adapted to respond to my major comment in methodology section about the analysis. The suggestion of a percentage of animals rehomed from the wider numbers kept in UK facilities has been removed. 

INTRODUCTION

Line 58: “The practice supports the notion that…”. I do not agree with “supports”. The practice of rehoming is guided by the notion that animals are sentient beings etc. Thank you, this has been modified accordingly. 

Lines 69-70: these arguments are worded too strong. They may be possible, but certainly not true for all cases. Please re-word. Re-worded to “Firstly, surplus animals have not been subject to research, and thus long-term health implications (often cited as a barrier to rehoming) are less likely to represent a risk [8]. Secondly, rehoming these animals typically presents a lower risk of disease transmission.”

Line 78-79: again too strong. In veterinary practice, euthanasia is also frowned upon many times. And routine euthanasia (which can happen e.g. in an overcrowded shelter) is never regarded as favourable. Thank you for this thoughtful consideration. This has now been changed to “Euthanasia may have more positive connotations in the veterinary clinic setting when companion animals are relieved of suffering, but in the animal laboratory, issues with routine euthanasia are compounded within a setting where animals are systemically harmed for human benefit.”

Line 91 and beyond: I am concerned about the authors’ understanding of the concept of animal welfare, because of the references they use in this section. The fact that they refer to a philosophical paper to define animal welfare concerns me. There are good papers by animal welfare scientists, which should be preferred as a source. See for example: Broom 2011, Acta Biother, 59:121-137 or Weary and Robbins 2019, Anim Welf, 28:33-40. An updated definition of animal welfare has been included, based on Weary and Robbins (2019) – “Animal welfare typically includes a consideration of animal’s affective state (pain), biological functioning (injuries) and sometimes also a consideration of naturalness (such as pasture access).” The definition of animal welfare from a philosophical paper has been removed. 

Line 130-142: I find this paragraph confusing. The authors could elaborate a bit on the results from the outcome studies, to indicate good practices in rehoming already. Thanks, this is a good suggestion. We have now also added data on the success of current rehoming schemes – “Research showed beagles have a high adaptive capacity, fit in well with families and thus make good companion animals [25]. Within 6-12 weeks of rehoming, behaviour tests on the dogs reflected relaxed body language, as well as reduced heart rates, signifying calm behaviour once settled in the home environment [25]. Research also found cats were successfully rehomed from a research facility, with a retention rate of 93.5% [23]. Of those rehomed, 80.4% were considered a valued family member, and behavioural problems were reported in just 11.3% of the cats within 6 months of adoption [23]. Research revealed that Cornell University developed a successful direct adoption scheme, and that the University of California, San Francisco employs an indirect scheme, whereby animals are transported to a third party shelter before rehoming [10]. Both schemes were judged to be successful, with “hundreds” of animals placed in adoptive homes [10]. These case studies convey hope that rehoming as a practice could be successfully institutionalised and adapted to specific facility needs, across a range of sentient species.”

Line 133: “The overwhelming focus on these species is limiting”: I do not understand what the authors are trying to say. We understand the confusion - the sentence has been adapted to be clearer – “However, the overwhelming focus on these species does not provide a comprehensive understanding of all rehoming practice across species” 

Line 134-137: The references 5 and 20 are books. Very unlikely that these report original research. Can primary sources be used? Or are the authors of the books referring to anecdotal evidence? This needs to be made more clear, because the authors state “as research points out…”, so the supporting references should be research papers. Thank you for pointing this out. We have removed as ‘research points out’, as the authors of the book are referring to anecdotal evidence. However, we cannot find any primary sources of research that can be used in this context. 

Line 148: “This is necessary to enhance animal welfare”: I find this quite a vague statement and would like to know what the authors mean. Is it about enhancing animal welfare by rehoming, or enhancing animal welfare by rehoming correctly (using proper procedures, which can help guarantee the welfare of the animals after rehoming, as demanded by law)? This has now been updated to reflect reviewer’s concerns to “This is necessary to enhance animal welfare by following correct rehoming procedures which can help to ensure animal welfare after rehoming, as stipulated by law, and promote the rehoming of laboratory animals where viable”

METHODOLOGY

The methods section needs to mention when the survey was administered (months & year). We have now provided information on when responses were collected: “Reponses were collected between July 2018 – January 2019.”

Line 178 and 242: “the university of XXXXX” – I assume the authors have blinded the university, but since this is not a blinded review, that is not necessary. Changed as suggested to include the University of Southampton. 

Line 179: “spilt” should be “split”. Thank you! This has been corrected.

Line 192: AWERB should appear in full the first time it is used. Thank you – we have added in an explanation of the abbreviation. 

Lines 192-195: this information should be presented at the beginning of the results section. The sentence “Participants represented a variety of roles, including but not limited to: Establishment Licence Holders (ELHs), Named Veterinary Surgeons (NVSs), AWERB (Animal Welfare and Ethical Review Body) chairs, Named Animal Care and Welfare Officers (NACWOs), and Named Information Officers (NIOs)” has been moved to beginning of results section. 

Line 216 and 496 (for example): sometimes GA is used and sometimes GM, while I think the authors mean the same. We have now used GA (in accordance with Home Office policy guidance) throughout the paper. 

Line 218: No need for this long sentence, just use the term that was introduced before: surplus animals. Thank you for the advice, we have removed the long sentence and used surplus. 

Lines 207-235: The data processing section is too long and fragmented. Also, crucial information is missing. Please create one or two paragraphs with data processing information, concisely reporting how the responses to the questions were processed (how were closed questions processed, what about open questions – was thematic analysis used?), which variables were calculated, why and how….Thank you for this helpful advice. The data processing section has now been cut back to two shorter paragraphs. More data has been provided regarding the way in which the open question data was analysed: “Analysing data from the open questions involved a structured inductive thematic analysis. This helped to identify common topics, ideas, concepts and patterns from the data. In order to do this, we used Nvivo12 to code the data and to generate themes from it. Using Nvivo12, the thematic analysis also involved a frequency count, where it was possible to see the number of times each identified theme was referenced across all participants”. More detail has also been provided regarding how closed question responses were analysed: “Nvivo12 was used to analyse the qualitative responses, Microsoft Excel 2016 to analyse quantitative findings and assist in numerical analysis (including calculating numbers of animals rehomed), and SigmaPlot (Version 13) to assist in graph production”. As we looked at the sample (41 facilities) as a whole, no independent variables were included. Further, the questionnaire involves descriptive statistics, so there is no need to explain how independent variable were calculated.

RESULTS

Lines 259-261: this is a bold statement that is insufficiently supported by the data that are presented. How does the variety in the survey relate to the variety in all other facilities? Thank you for the advice – this statement has been removed. 

Figure 1: I would delete figure 1. There is no added value there, as the numbers are already in the text. Thanks – Figure 1 has been deleted in line with reviewer comments. 

Line 278-280: I find this an odd statement, given that, equally, it is very possible institutions that do not rehome did not complete the survey. Also, does this mean that incomplete surveys were not counted? This needs to be stated in the data processing part of the Methodology section. We have now deleted “It is possible that facilities currently engaged in rehoming did not complete the questionnaire, nevertheless 19 represents more than 1 in 10 research facilities.” We have also added into the methodology that incomplete surveys were not counted.

Line 289: “If we assume that the ratio above extends” – to what? This section of the sentence has been removed, as it was unclear.

Line 291: This is an example of how minimum percentage is not used throughout. Because it is a minimum, you can only state that “at least 0.02% of laboratory animals are known to be rehomed from 2015-2017”, but again, this figure to me is really misleading as it is a huge underestimation. Thank you - this figure has been removed throughout the paper in line with wider comments regarding the analysis. 

Line 300: keep in mind that primates may not be rehomed in the UK sense of the word, but they may be retired from research, while still living at the facility. Thanks – we have added in this thoughtful contribution. 

Table 1: again, misleading due to the “known percentage rehomed” calculation. Similar comment for the last column with the ration. This has been removed from the table. 

Figure 4: The left pie chart is difficult to read. I would turn this figure into a table comparing proportions of animals rehomed (41 institutions) and animals kept (UK total). This, in my opinion is the only valid numerical comparison that could be made (see my major concern about the calculation of “known percentage rehomed”). Thank you for your advice, the pie chart has now been converted to a table.

Lines 349 and beyond: the presentation of results is very qualitative as it almost looks at individual responses from respondents. I feel a more structured analysis, like a thematic analysis with a frequency count, is appropriate. This is a very useful contribution, a structured analysis with a frequency count has now been included. 

Lines 487-489: The grouping of reasons for not rehoming does not adequately capture the responses. E.g. Fear of unwanted or negative media attention is not a practical issue. Please re-consider this. We have amended this to include three groupings; “welfare concerns with regard to the animal’s health if rehomed, practical issues surrounding demand and the fact that research needs’ tend to leave few animals to retire, and external issues including fear of negative media attention”.

DISCUSSION

Lines 526-527: “this is also demonstrated by only one facility of the 41 facilities”: it is not clear what is demonstrated. This sentence has been made clearer – it is the fact rehoming is often a consideration in UK research facilities that is demonstrated. 

Line 545 and beyond: the authors seem to suggest that all GM models pose a biosecurity risk, which is not true. Added in “Despite it being scientifically unclear exactly how some GA animals might pose a biosecurity risk, the rehoming of GA animals was considered illegal by some who responded to the survey.”

Line 585: “this effect” – which effect? Please state clearly to what you are referring. Thank you for your advice, this has now been made clearer – the effect referred to is the improvement in staff wellbeing through rehoming.

---

## [Decision Letter · Decision Letter 2]

1 Jun 2020

PONE-D-20-01484R2

A semi-structured questionnaire survey of laboratory animal rehoming practice across 41 UK animal research facilities

PLOS ONE

Dear Dr. Skidmore,

Thank you for submitting your manuscript to PLOS ONE. After careful consideration, we feel that it has merit but does not fully meet PLOS ONE’s publication criteria as it currently stands. Therefore, we invite you to submit a revised version of the manuscript that addresses the points raised during the review process.

I recognize that you have addressed most of the concerns raised in previous rounds of review, and that the remaining issues are fairly minor. In addition to the reviewers' comments, there are also a few arising from my own thorough reading of the current version of your submission. You will find all of these below.

We look forward to receiving your revised manuscript.

Kind regards,

I Anna S Olsson, Ph.D.

Academic Editor

PLOS ONE

Additional Editor Comments (if provided):

General: It is debatable whether the term euthanasia should be used when animals are killed in situations when it is not in their own interest. I recognize that the term euthanasia is predominant when referring to animals killed for research purposes, but I invite you to reflect on whether it is the best term in your paper. For example, you write "Euthanasia is undertaken for three primary reasons – 1) as a scientific  requirement, 2) to prevent avoidable suffering, or 3) for financial/logistical reasons". Critics would say that the term "euthanasia" only applies to 2) in the cited example. I understand that you are aware of the controversy and briefly mention it in the introduction with a reference to Kuře, J., Euthanasia: The" Good Death" Controversy in Humans and Animals. 2011: BoD–

652 Books on Demand. Please consider changing to "killing" and justify your choice.

Table 1 I really appreciate the use of colour coding - please add that higher numbers are represented in more saturated colours.

Line 379 the owner is provided with appropriate housing and feeding **for the rehomed animal(s)**.

Line 155 Existing literature notes this as **an** essential next step in (or **the** essential)

Lines 474-476 "there are challenges that come with rehoming to staff – an NVS explained how the rodents could not be rehomed to employees as staff might “acquire rodents from other sources” that are “microbiologically dirty […] which could present a risk of inadvertent delivery of disease”. This statement is a little puzzling as there is no obvious relation between rehoming rodents to staff and the same staff acquiring rodents from other sources. I assume that the relation may be that because microbiological contamination is considered a risk, staff are not allowed to keep rodents as pets, and therefore can't receive rehomed rodents. Do you know if this is the case?

Lines 496-500 and section Biosecurity Please provide a reference for that rehoming GA animals is not in itself illegal in the UK.

Figures 3-5: Please consider merging the 3 sets of graphs into one figure with 3 panels.

Reference list: Reference manager generated lists require careful manual checking. For example, "Office, H."  is not the right way to refer to the Home Office! There are also inconsistencies in whether journal titles are capitalized or not (they should be).

Reviewers' comments:

Reviewer's Responses to Questions

**Comments to the Author**

1. If the authors have adequately addressed your comments raised in a previous round of review and you feel that this manuscript is now acceptable for publication, you may indicate that here to bypass the “Comments to the Author” section, enter your conflict of interest statement in the “Confidential to Editor” section, and submit your "Accept" recommendation.

Reviewer #1: All comments have been addressed

Reviewer #2: (No Response)

2. Is the manuscript technically sound, and do the data support the conclusions?

Reviewer #1: Yes

Reviewer #2: Yes

3. Has the statistical analysis been performed appropriately and rigorously? 

Reviewer #1: Yes

Reviewer #2: N/A

4. Have the authors made all data underlying the findings in their manuscript fully available?

Reviewer #1: Yes

Reviewer #2: No

5. Is the manuscript presented in an intelligible fashion and written in standard English?

Reviewer #1: Yes

Reviewer #2: Yes

6. Review Comments to the Author

Reviewer #1: Thank you for this very careful revision. You have implemented the reviewers' comments well. Please capitalize the word "owner" in the captions of Tables 4 and 5.

Reviewer #2: The authors were able to significantly improve the manuscript.

There is a remaining major concern about the discussion for me, which requires some additional work and consists of the following:

- Although it is true that not many papers exist on rehoming of research animals, there are some. Particularly with relevance to the section "finding the right home", I feel that it is approporiate to compare the practices to the shelter literature and investigated approach / recommended good practices.

- The authors report on the involvement of third party rehoming organisations. However, they should at least consider that these consist of volunteers who mean well, but who are rarely trained professionally for the duties they perform. This should be briefly discussed in relation to available literature on animal welfare volunteers.

- The fact that an extrapolation was made from 2017 data to 2015 and 2016 (see lines 215-217 in the methods section) should be mentioned as a limitation to the study, as this is based on the assumption that numbers are similar in those three years (if there is a good reason to assume this, then it should be mentioned as well).

The remaining minor concerns:

Line 37: "this period" should be clarified

Line 104: "desire" is perhaps a bit inappropriately anthropomorphic. Perhaps better to say something like "includes the degree to which the animal is provided with what it needs as well as wants" (or use "needs and wants" as these are commonly used terms in animal welfare science)

Line 142: why use "sentient" here explicitly for companion animals? This implies that the subsequent species mentioned (rabbits, rats, guinea pigs and mice) are not sentient, which is not commonly accepted as they are vertrebrates as well.

Table 1 : mention the number of UK facilities in the column header of the second column. In the caption, explain the coding of the shading, i.e. darker shading indicates a higher number.

Lines 300 - 314: this feels repetitive (title is also almost identical to the one above). Why not introduce the percentages of animal groups in the text in the section above, and then refer to table 1 for details?

Tables 3 and 4: captions should mention the total number of respondents

Lines 378-379: did the institution provide all the housing and food that the animal needed or did they provide some of it? It would be useful to have a specific example of what is meant.

Lines 568-571: sentence not entirely clear, particularly the first part "this does not mean rehoming... kept in UK facilities".

7. PLOS authors have the option to publish the peer review history of their article (what does this mean?). If published, this will include your full peer review and any attached files.

Reviewer #1: No

Reviewer #2: No

---

## [Author Response · Author response to Decision Letter 2]

3 Jun 2020

Thank you all for your very helpful comments. 

General: It is debatable whether the term euthanasia should be used when animals are killed in situations when it is not in their own interest. I recognize that the term euthanasia is predominant when referring to animals killed for research purposes, but I invite you to reflect on whether it is the best term in your paper. For example, you write "Euthanasia is undertaken for three primary reasons – 1) as a scientific requirement, 2) to prevent avoidable suffering, or 3) for financial/logistical reasons". Critics would say that the term "euthanasia" only applies to 2) in the cited example. I understand that you are aware of the controversy and briefly mention it in the introduction with a reference to Kuře, J., Euthanasia: The" Good Death" Controversy in Humans and Animals. 2011: BoD–652 Books on Demand. Please consider changing to "killing" and justify your choice. Thank you for your suggestion – this is really interesting. I have now used the word ‘killing’ throughout, but have explained the term euthanasia to relate to the act of putting an animal out of avoidable suffering. 

Table 1 I really appreciate the use of colour coding - please add that higher numbers are represented in more saturated colours. Thank you, I have now added this in. 

Line 379 the owner is provided with appropriate housing and feeding for the rehomed animal(s). Thank you, I have changed in line with your comments. 

Line 155 Existing literature notes this as an essential next step in (or the essential). Thank you! This has been changed accordingly. 

Lines 474-476 "there are challenges that come with rehoming to staff – an NVS explained how the rodents could not be rehomed to employees as staff might “acquire rodents from other sources” that are “microbiologically dirty […] which could present a risk of inadvertent delivery of disease”. This statement is a little puzzling as there is no obvious relation between rehoming rodents to staff and the same staff acquiring rodents from other sources. I assume that the relation may be that because microbiological contamination is considered a risk, staff are not allowed to keep rodents as pets, and therefore can't receive rehomed rodents. Do you know if this is the case? Yes – that is the case. I have now made this clearer, and inserted: “Microbiological contamination is thus considered a risk, and therefore some facility staff cannot keep rodents as pets, including those from the laboratory.”

Lines 496-500 and section Biosecurity Please provide a reference for that rehoming GA animals is not in itself illegal in the UK. Thank you, I have now inserted a reference for the Home Office 2015 Advice Note that outlines that additional legislation may also apply to the rehoming of genetically altered animals. 

Figures 3-5: Please consider merging the 3 sets of graphs into one figure with 3 panels. Thank you for the useful suggestion, I have now merged the 3 graphs into one figure. 

Reference list: Reference manager generated lists require careful manual checking. For example, "Office, H." is not the right way to refer to the Home Office! There are also inconsistencies in whether journal titles are capitalized or not (they should be). Thanks - this has now been addressed. 

Reviewer #1: Thank you for this very careful revision. You have implemented the reviewers' comments well. Please capitalize the word "owner" in the captions of Tables 4 and 5. Thank you for your very helpful comments – tables 4 and 5 now have a capitalised ‘Owner’.

Reviewer #2: The authors were able to significantly improve the manuscript.

There is a remaining major concern about the discussion for me, which requires some additional work and consists of the following:

- Although it is true that not many papers exist on rehoming of research animals, there are some. Particularly with relevance to the section "finding the right home", I feel that it is appropriate to compare the practices to the shelter literature and investigated approach / recommended good practices. Thank you for this very useful suggestion. I have added in some animal shelter literature and compared this to the findings of this research, mainly under the section “finding the right home”. 

- The authors report on the involvement of third party rehoming organisations. However, they should at least consider that these consist of volunteers who mean well, but who are rarely trained professionally for the duties they perform. This should be briefly discussed in relation to available literature on animal welfare volunteers. Thanks for this – I have now reflected on this: “However, some of those that work at rehoming organisations are volunteers – and literature notes that these volunteers are not always taught the required skills to adequately train and socialise animals [32] In fact, only 12% of animal shelter employees across the US rated volunteers as well trained “to a great extent” [33]. Research also finds an irregular schedule of social contact with animals from volunteers and frequent changes in active volunteers [34, 35].”

- The fact that an extrapolation was made from 2017 data to 2015 and 2016 (see lines 215-217 in the methods section) should be mentioned as a limitation to the study, as this is based on the assumption that numbers are similar in those three years (if there is a good reason to assume this, then it should be mentioned as well). Thank you for this useful thought – I have now added “Finally, an extrapolation was made from 2017 Home Office data to the years 2015 and 2016 to calculate an approximate number of surplus animals across the 3 years. However, there is likely to be some variance in the proportions of surplus animals by species year by year that is not accounted for.”

The remaining minor concerns:

Line 37: "this period" should be clarified. Thank you, this period has now been clarified to between 2015-2017.

Line 104: "desire" is perhaps a bit inappropriately anthropomorphic. Perhaps better to say something like "includes the degree to which the animal is provided with what it needs as well as wants" (or use "needs and wants" as these are commonly used terms in animal welfare science). I have now added in “It includes the degree to which the animal is provided with its needs and wants”.

Line 142: why use "sentient" here explicitly for companion animals? This implies that the subsequent species mentioned (rabbits, rats, guinea pigs and mice) are not sentient, which is not commonly accepted as they are vertebrates as well. I have now removed the word sentient.

Table 1 : mention the number of UK facilities in the column header of the second column. In the caption, explain the coding of the shading, i.e. darker shading indicates a higher number. Thank you for your comments, I have now added in an explanation of the darker shading, and the total number of UK facilities. 

Lines 300 - 314: this feels repetitive (title is also almost identical to the one above). Why not introduce the percentages of animal groups in the text in the section above, and then refer to table 1 for details? Thank you – I have removed table 2, and added in the text: “Although 94.15% of species kept in laboratories are rodents, they make up under a fifth (19.14%) of all animals known to be rehomed between 2015-2017. Conversely, birds, cats, dogs, horses, amphibians and agricultural animals constitute 80.86% of total species rehomed, despite making up just 5.84% of those kept (see Table 1 for more details). This is based on the following grouping: dogs (beagles and all other dog breeds), small mammals (rats, mice, gerbils, rabbits, hamsters, ferrets, guinea pigs), birds (common quail and all other birds), agricultural animals (cattle, sheep, pigs and goats), and cats, horses, amphibians and primates. Fish were excluded from this analysis due to one outlier facility having rehomed over 1200. There thus exists a preference for the rehoming of some species over others”

Tables 3 and 4: captions should mention the total number of respondents. I have now added in n=19 on the table caption. 

Lines 378-379: did the institution provide all the housing and food that the animal needed or did they provide some of it? It would be useful to have a specific example of what is meant. Thank you – I have explained that it was generally initial food to help the animals acclimatise. I have also added in a quote “The animals are always released with items from their home cages and some diet. We inform the new owners it will take them quite a while to acclimatise and for the initial few days just to place their hands in the cage and let the animals come to them” to provide a specific example of what is meant.

Lines 568-571: sentence not entirely clear, particularly the first part "this does not mean rehoming... kept in UK facilities". Thank you – this had been modified to “Despite the clear impossibility of finding homes for all of the rodents and fish currently kept in UK facilities, there would be merit in understanding the drivers and processes of rehoming even small numbers of these species.”

---

## [Editor Report · Decision Letter 3]

5 Jun 2020

A semi-structured questionnaire survey of laboratory animal rehoming practice across 41 UK animal research facilities

PONE-D-20-01484R3

Dear Dr. Skidmore,

We’re pleased to inform you that your manuscript has been judged scientifically suitable for publication and will be formally accepted for publication once it meets all outstanding technical requirements.

Kind regards,

I Anna S Olsson, Ph.D.

Academic Editor

PLOS ONE
---

## [Editor Report · Acceptance letter]

9 Jun 2020

PONE-D-20-01484R3 

A semi-structured questionnaire survey of laboratory animal rehoming practice across 41 UK animal research facilities 

Dear Dr. Skidmore:

I'm pleased to inform you that your manuscript has been deemed suitable for publication in PLOS ONE. Congratulations! Your manuscript is now with our production department. 

Kind regards, 

on behalf of

Dr I Anna S Olsson 

Academic Editor

PLOS ONE